# Process- and Product-Related Foulants in Virus Filtration

**DOI:** 10.3390/bioengineering9040155

**Published:** 2022-04-04

**Authors:** Solomon Isu, Xianghong Qian, Andrew L. Zydney, S. Ranil Wickramasinghe

**Affiliations:** 1Ralph E. Martin Department of Chemical Engineering, University of Arkansas, Fayetteville, AR 72701, USA; soisu@uark.edu; 2Department of Biomedical Engineering, University of Arkansas, Fayetteville, AR 72701, USA; xqian@uark.edu; 3Department of Chemical Engineering, Pennsylvania State University, University Park, PA 16802, USA; zydney@engr.psu.edu

**Keywords:** aggregation, charge variant, downstream processing, fusion protein, foulant, glycoform, prefiltration, process development, monoclonal antibody, virus filtration

## Abstract

Regulatory authorities place stringent guidelines on the removal of contaminants during the manufacture of biopharmaceutical products. Monoclonal antibodies, Fc-fusion proteins, and other mammalian cell-derived biotherapeutics are heterogeneous molecules that are validated based on the production process and not on molecular homogeneity. Validation of clearance of potential contamination by viruses is a major challenge during the downstream purification of these therapeutics. Virus filtration is a single-use, size-based separation process in which the contaminating virus particles are retained while the therapeutic molecules pass through the membrane pores. Virus filtration is routinely used as part of the overall virus clearance strategy. Compromised performance of virus filters due to membrane fouling, low throughput and reduced viral clearance, is of considerable industrial significance and is frequently a major challenge. This review shows how components generated during cell culture, contaminants, and product variants can affect virus filtration of mammalian cell-derived biologics. Cell culture-derived foulants include host cell proteins, proteases, and endotoxins. We also provide mitigation measures for each potential foulant.

## 1. Introduction

A virus filtration step is frequently included to provide a robust size-based clearance of both enveloped and non-enveloped viruses during the manufacture of mammalian cell-derived biotherapeutics, such as monoclonal antibodies (mAbs) and Fc-fusion proteins [1,2]. Before approval of new therapeutics, regulatory agencies such as the US Food and Drug Administration (FDA) require validation of adequate virus clearance [3]. Consequently, unit operations are added to the purification train to ensure high levels of virus clearance [4]. Virus filtration uses large pore size ultrafiltration membranes to retain any contaminating virus particles while recovering the virus-free product in the permeate. Unlike conventional ultrafiltration operations, the performance criteria for virus filters are far stricter [5]. Typically, around 95% product recovery is required while maintaining at least 1000 fold (3 log reduction) virus clearance [2].

Table 1 lists a range of mammalian cell-derived biotherapeutics that have been approved by the Food and Drug Administration (FDA) over the last three decades. The monoclonal antibody industry sector grossed over USD 154 billion in 2020 [6,7]. Mammalian cells used for expression of recent FDA-approved monoclonal antibodies include Chinese Hamster Ovary (CHO) cells and murine myeloma cells (Sp2/0, NS0), among others [6,8].

Virus filtration is different from typical pressure-driven membrane filtration processes, as the filter is designed to obtain very high levels of removal of potential virus contaminants. Further, as it is impractical to validate that there is zero carryover of any trapped virus particles, reuse of the virus filter is impossible. Consequently, these are single-use devices. Virus filters are typically run in normal flow (dead end) mode, rather than tangential flow mode used for protein ultrafiltration, since normal flow is less complex and requires only a single pump.

The performance of virus filters is measured in terms of product recovery, log reduction value (LRV) of the virus (defined as the logarithm to base 10 of the ratio of the virus concentration in the feed to that in the permeate), and the productivity of the filter. Productivity is typically expressed as the volume of feed that can be processed per membrane area (L·m^−2^) before the filtrate flux has decreased to unacceptably low levels (for operation at constant transmembrane pressure). Since biopharmaceutical manufacturing operations are still essentially batch processes, the virus filter is often sized such that the entire batch can be processed in one shift.

Frequently, identifying a virus filter that meets the three performance requirements, product recovery, LRV, and productivity, is challenging and highly dependent on the feed stream and membrane properties. As the virus filtration step is located towards the end of the purification train, the product is highly purified and moderately concentrated [2]. Membrane fouling, which leads to compromised performance, is typically due to product- and process-related foulants rather than any rejected virus particles [2], since the concentration of virus particles in any process will be orders of magnitude less than that for the product.

In order to remove impurities and foulants, virus filtration membranes are sometimes designed with a reverse asymmetric structure [10]. In this case, the barrier layer faces away (downstream) while the more open support layer faces towards the feed stream [11]. The support layer can act as an inline prefilter that traps larger foulants and protects the tight barrier layer [12]. However, essentially symmetric membranes are also used industrially. The unique requirements of virus filtration are very different from typical pressure-driven membrane separation processes such as ultrafiltration. Identifying and sizing an appropriate virus filter is often particularly challenging.

This review describes a typical ‘platform process’ for the downstream purification of biopharmaceutical products. First, the location of the virus filtration step in the downstream processing workflow is identified. Next, the major commercially available virus filters are summarized. The remaining sections of this review highlight various impurities and foulants that could lead to fouling and compromised performance during virus filtration of mammalian cell-derived biologics. Inline virus prefilters that are frequently used to remove product-related aggregates are also discussed. The review ends with a discussion of future trends in the development of virus filters.

## 2. Downstream Processing

### 2.1. Platform Processes

Biopharmaceutical manufacturing processes can be divided into two main processing trains: upstream cell culture operations and downstream purification processes. Various bioreactor configurations are used to produce the cells that express the product of interest (mAbs, enzymes, Fc-fusion proteins, or hormones). Removing particulate matter such as cells and cell debris occurs at the interface between upstream and downstream unit operations. These bioreactor clarification operations are sometimes referred to as midstream processes [13,14].

Figure 1 is a typical ‘platform’ process for the downstream purification of monoclonal antibodies. The first unit operation is typically an affinity chromatography capture step using protein A (resin-based chromatography) [15]. Affinity interaction is a specific interaction based on both the topological fit and a combination of electrostatic, hydrophobic, and hydrogen-bonding interactions [16]. Antibody elution from the protein A column is performed at low pH, making it very convenient to include a low pH hold for virus inactivation.

Frequently, two polishing steps are used to remove the remaining impurities and product variants/aggregates [17]. Resin- or membrane-based chromatography (ion exchange or hydrophobic interaction chromatography) is frequently used. The polishing steps remove impurities such as DNA, host cell proteins (HCP), and product aggregates [2]. Typically, all streams and buffers which enter the purification process are passed through sterilizing grade (0.22 μm pore size) filters to reduce bioburden.

As shown in Figure 1, the virus filtration step is typically located near the end of the purification train. The product is relatively concentrated and highly purified. High product concentrations can lead to compromised performance due to product aggregation and increased adsorption to the virus filter membrane. A final ultrafiltration/diafiltration step is used to concentrate the product and place it in the formulation buffer needed for stability during shipping/storage and delivery to the patient. The final 0.22 µm pore size filter is used to ensure sterility of the product and is often part of the final fill-finish operation.

### 2.2. Viruses, Virus Clearance, and Virus Filters

Many mammalian cell lines produce endogenous retrovirus-like particles [2]. These particles are typically around 80–100 nm in size. Clearance can be achieved by inactivation and/or physical removal from the process stream [12,18,19]. During purification, manufacturers of mammalian cell-derived biotherapeutics must demonstrate that the process will yield a final product containing no more than one virus particle in a million doses. Estimates of the number of virus particles in a single dose equivalent from the bioreactor could be as high as 10^10^–10^15^ retrovirus-like particles per mL [2]. Removal of adventitious viruses such as parvovirus is also required. These much smaller viruses are around 20 nm in size. In the past, filters targeted for retrovirus and parvovirus removal were included in the purification train [3]. Recent studies show that virus clearance filters designed to provide clearance of smaller parvovirus can be used to clear much larger retroviruses simultaneously [20].

Table 2 shows some viruses that are employed for validation studies in biomanufacturing. The enveloped retroviruses are typically larger than the non-enveloped parvoviruses. Consequently, virus filtration membranes that are validated for removal of parvovirus are also effective at clearing retrovirus from the product.

Adventitious virus contamination is a concern in the manufacture of biologics. Validation of virus clearance is shown by conducting scale-down testing [22]. The feed is spiked with model virus particles, and clearance in the product stream is determined. Minute virus of mice (MVM, mouse parvovirus) is often used to validate adventitious virus clearance. The FDA requires at least two orthogonal steps with different mechanisms of action for validation of virus clearance with the required level of virus clearance for the process as a whole, determined by summing the clearances obtained from the individual unit operations [2].

Virus filtration uses porous polymeric membranes in normal flow mode [12,23,24]. The predominant mechanism of action for virus filters is size exclusion [12]. The difference in hydrodynamic diameter between a protein product and MVM is often less than two-fold [5]. Today, virus filters are a critical component of the overall virus clearance strategy [12]. As shown in Table 3, virus filter membranes are typically made of regenerated cellulose, polyvinylidene difluoride (PVDF), and polyethersulfone. The latter two materials are hydrophilized in order to minimize fouling by adsorption and maximize flux during virus filtration. While the membrane should be biocompatible, non-fouling, and minimize adsorption on the membrane surface, it is also essential that the membrane is robust and dimensionally stable to ensure the required level of virus clearance.

These virus filters are designed to ensure that only monomeric biomolecules with a hydrodynamic diameter less than 20 nm can pass through the pores. Much research is needed to understand how a multidomain, anisotropic mAb with varied surface moieties interacts with virus filtration membranes, prefilters, and other product monomers [25].

Table 3 is a non-exhaustive list showing commercially available virus filters and material configurations. Operating pressures and permeate fluxes vary greatly. Virus filter membrane fouling is a significant challenge [24,28,29]. Fouling can compromise virus clearance and reduce membrane productivity (product recovered per membrane surface area) [30]. Fouling is often due to product variants because of the high product purity before virus filtration and the high product concentration compared to the spiked virus concentration [2].

Recent studies focusing on virus filtration of mAbs showed that membrane performance depends on the mAb properties (pI, hydrophobicity, net charge, dipole moment, oligomericity), buffer conditions, membrane material, and operating pressure [24,31]. Buffer excipients such as arginine and lysine can stabilize mAbs and reduce fouling propensities [32]. Excipients such as histidine, arginine, and lysine can reduce reversible self-association of mAbs to varying degrees [33]. Reversible self-association is often concentration-dependent [33].

## 3. Virus Filter Foulants

This section describes the major classes of foulants in virus filtration. This includes irreversible and reversible product aggregates and minor product variants that differ in their charge or hydrophobicity. Product variants arise because mammalian cell-derived biotherapeutics are heterogeneous. The product is defined based on the production process and not on a single molecular species. Product variants with different post-translational modifications can have different hydrophobicity, charge, and conformations. If present, HCP, proteases, and nucleic acids can also foul the virus filter.

### 3.1. Monoclonal Antibody Aggregates

Aggregation is a typical occurrence with mAbs and other therapeutic proteins. Several pathways have been proposed to describe the aggregation of proteins. They include agglomeration of monomers in their native states, aggregation of conformationally altered or chemically modified monomers, nucleation, and surface-induced aggregation [34,35,36]. Significant attention has been placed on non-native monomer aggregation, since exposed hydrophobic moieties tend to self-associate [34]. Some surfactants, osmolytes, and chaotropes induce aggregation because they denature the monomeric product, exposing more of the hydrophobic core and distorting the surface charge distribution [37]. Physical and biochemical events can also induce product degradation through enzymatic and non-enzymatic processes such as shock, light, and oxidation [38].

Physical or chemical perturbations that put a strain on the native conformation of biotherapeutic proteins such as mAbs can result in clipping or aggregation [39]. Such conditions include the presence of chaotropic chemical species, pH swings [40], shock, mechanical stress, increased concentration, and large temperature fluctuation [34,41,42]. The size, charge, and hydrophobicity of a mAb aggregate will differ from that of the native structure.

Interfacial damage can also affect the stability of a product monomer, especially at the air–liquid interface, which induces nucleation and aggregation [43]. Surface tension and physical adsorption on solid surfaces also lead to conformational changes [34,44,45]. Freezing and thawing of a product induces more aggressive fouling of virus filters [2,23,46]. Freeze–thaw-induced aggregation is due to conformational changes at the ice–water interface and by freeze concentration [34,44,47].

Reversible aggregates are usually a precursor to nucleation [48], followed by irreversible aggregation as the aggregates increase in size [49,50]. As buffer ionic strength increases, electrostatic repulsion between the mAbs decreases, whereas hydrophobic attraction between the mAb increases, often leading to product aggregation [44]. Aggregation and precipitation occur most easily at the product’s isoelectric point (pI) due to reduced electrostatic repulsion between individual product molecules [51,52].

#### 3.1.1. Reversible Aggregates

Aggregation can occur through different pathways, resulting in aggregates that are reversible or irreversible [34,37]. mAb oligomers such as dimers, trimers, and tetramers are typically reversible [34,35]. Reversible aggregates are known as soluble aggregates, and the associated product monomers are not significantly denatured. Soluble aggregates are caused by interactions between product molecules via hydrogen bonding, electrostatic, or van der Waals forces [37,53]. These soluble aggregates can foul virus filters if their size exceeds the 20 nm size cut-off of most parvovirus filters.

Rayfield et al. [28] investigated the impact of mAb properties on virus filter filterability and showed that aggregates bigger than 17 nm were correlated to the flux decline during virus filtration [2,28]. Monoclonal antibodies typically have a hydrodynamic diameter of 9–12 nm; thus, the small oligomers can be 20 nm to as much as 50 nm in diameter. Other studies have shown that freeze-thawing of mAbs may not cause aggregation in significant amounts detectable by size exclusion chromatography due to the relatively small diameters of potential aggregates formed [46,54,55].

#### 3.1.2. Irreversible Aggregates

When product dimers and trimers undergo further aggregation, they attain a critical mass where the aggregate can no longer remain soluble. These large aggregates then precipitate out of the solution [34]. The precipitates become visible and show increased turbidity and cloudiness. These large aggregates are known as irreversible aggregates. Large, insoluble aggregates have an increased propensity to foul the separation-active layer during virus filtration. Barnard et al. investigated the principal foulant of freeze–thawed mAb solutions and found that the freeze–thaw process could induce the formation of large aggregates (>1 μm) [46]. The use of 0.1 or 0.22 µm pore size prefilters can mitigate virus filter fouling to some extent by removing these large aggregates.

Irreversible aggregation is prevalent with denatured product monomers [35,37]. Chemical degradation, such as oxidation and deamidation, alters the surface charge of product monomers and affects colloidal stability [35]. The irreversible aggregation of a product results in product loss, although even very low levels of aggregation (<1%) can cause a significant increase in filter fouling. Hawe et al. studied mAb aggregates formed during freeze–thaw- and heat-induced thermal stress [45]. Other studies show that heat denatures mAbs and leads to irreversible mAb aggregation [35,37].

### 3.2. Host Cell Proteins (HCP), Proteases, and Nucleic Acids

HCPs feature significantly in the downstream processing of protein-based therapeutics [31]. HCPs include proteins, enzymes, and co-enzymes which emanate from the host cell used for product expression [56]. It is essential to remove HCPs from therapeutic proteins because they can elicit an immune response. There are regulatory requirements for robust HCP removal before clinical trials of drug candidates to prevent the development of anti-CHO antibodies by volunteers [57,58]. Some HCPs can co-elute with the mAbs through polishing and purification steps, either due to binding to the resin or to association with the mAb product [56,59,60,61]. Zhang et al. identified over 500 HCPs in a cell culture sample and tracked their fate through downstream processing unit operations [62]. After studying nine different mAbs, they determined that actin and clusterin were most abundant in protein A eluates [62].

Enzymatic HCPs (host cell proteases) can clip or denature product monomers, expose hydrophobic residues and charged moieties, and alter the product’s biophysical properties. Denatured product monomers with exposed residues induce virus filter fouling by adsorptive processes in addition to mAb–mAb and mAb–HCP association. Host cell proteases have been reported to result in the fragmentation of mAb products, with increased susceptibility to nucleation and aggregation [63]. However, proteases themselves are probably not principal foulants of virus filters, since virus filtration occurs towards the end of downstream processing, where only trace amounts of non-mAb impurities may be detected [19,64].

HCP diminishes the biotherapeutic quality of biotherapeutic products and increases downstream processing costs. If HCPs are not sufficiently removed, they could potentially induce flux decay during virus filtration. HCPs have a range of biophysical properties, such as pI (2–11) and mass (10–200 kDa), which can be used to separate the HCP from the biotherapeutic [65,66]. Protein A chromatography significantly reduces HCPs in the mAb product due to high selectivity for the Fc region of mAbs [59]. Several studies reported that the propensity of different HCPs to bind and co-elute with mAbs from protein A columns vary from mAb to mAb [60,67]. Problematic HCPs are many and include lipoprotein lipase, nidogen-1, clusterin, histones, keratins, phospholipases, ribosomal proteins, and serine proteases [65].

### 3.3. Endotoxins

Endotoxins or lipopolysaccharides (LPS) are contaminants that can enter the process through growth media or other cell culture additives used in mammalian cell cultures. LPS are produced by Gram-negative bacteria, commonly used in recombinant DNA production [68,69,70]. Endotoxins are commonly found contaminants in mammalian cell-derived therapeutics [71]. LPS are complex molecular conjugates of an amphiphilic component (lipid A) and a polar polysaccharide component [69,70]. The isoelectric point of LPS ranges from 1 to 4 [66]. LPS removal techniques that have been reported include two-phase extraction, affinity chromatography, and ion exchange chromatography [70].

LPS have been reported to have a high affinity for some biotherapeutic proteins [71]. LPS and therapeutic proteins can form micellar aggregates, complicating the removal process and potentially carrying over into the virus filtration step [70,72]. Phosphorylated moieties of LPS electrostatically bind with the carboxyl moieties of amino acids in the biologic of interest [70,72]. Solutions of 0.5 M arginine have been shown to promote LPS clearance during polishing steps [72].

LPS have molecular masses ranging from 3–40 kDa, which vary due to their polysaccharide chain lengths [66,70]. Endotoxins can co-elute with mAbs and Fc-fusion proteins from protein A resins by molecular conjugation through hydrophobic and electrostatic interactions, ultimately causing problems during virus filtration. Endotoxin-contaminated mAb streams have an increased propensity to cause virus filter fouling. Removal of endotoxins through ion exchange polishing steps increases the virus filtration capacity of virus filters.

### 3.4. Product-Mediated Foulants

#### 3.4.1. Charge Variants

The charge variant profile is a critical quality attribute of mAbs [73] and Fc-fusion proteins. Charge variants in mAbs can result from post-translational modifications (PTMs), such as deamidation of asparagine, C-terminal lysine variants, and glycosylation [74,75,76,77]. Glycans are mostly polar, hydrophilic oligosaccharides that can induce micro-differences in the surface charge of a glycoprotein. Negatively-charged glycans incorporating phosphorylated mannose and sialic acid can introduce micro-heterogeneities. Charge heterogeneity is observed in isoelectric-focusing electropherograms of most glycoproteins [74]. Acidic and basic variants of glycoproteins such as mAbs and Fc-fusion proteins can have different glycan profiles [78,79]. Meyer et al. [72] reported that specific charge variants of a mAb candidate were aggregation prone. Acidic variants of this mAb showed more pronounced hydrophobicity [80].

The net charge and surface charge distribution of glycoproteins change with buffer pH [76,81,82]. The pI of a protein is the pH value at which the net charge is zero [51]. For most mAbs, the pI ranges from 6.5–9.5 [51]. There is more biochemical variability with Fc-fusion proteins. A protein will be net negatively charged when the buffer pH is above the pI and positively charged when the buffer pH is below the pI [83]. Exposed surface residues on a glycoprotein can become protonated or deprotonated depending on the buffer pH, thereby inducing localized charged groups [84]. Charged moieties due to glycosylation, phosphorylation, and other PTMs affect the net charge of glycoproteins and their interactions with other product monomers and virus filtration membranes [83].

#### 3.4.2. Denatured Variants

Hydrophobic interaction is the preferential association of non-polar residues in aqueous media [85]. Amino acids with non-polar side chains are typically hydrophobic, e.g., valine, leucine, proline, and tryptophan. Polar amino acids such as arginine impart hydrophilic attributes to glycoproteins [86]. When hydrophobic amino acids are surface-exposed on a glycoprotein, hydrophobicity increases. Hydrophobic amino acids tend to be buried in the globular core of most glycoproteins. The hydrophobicity of a protein is also affected by the buffer pH and the protein’s charge state [87]. When the buffer pH is close to the pI of the protein, the protein is the most hydrophobic [85]. Denaturation and unfolding of glycoproteins can lead to variants with a higher fouling propensity on virus filters.

The glycan appendages of glycoproteins also contribute to the final stable conformation, and glycan variation can introduce minor hydrophobicity variations. Careful handling and mild changes in formulation conditions will reduce the formation of conformational variants, which could foul virus filters or induce product aggregation.

#### 3.4.3. Sequence Variants

Monoclonal antibodies and Fc-fusion proteins consist of amino acids in specific sequences that form secondary, tertiary, and quaternary structures. Sequence variants arise due to genetically unprogrammed amino acid substitutions, omissions, or insertions during biosynthesis [88]. Sequence variants result in macro-heterogeneities with biomolecular differences from the desired product [88]. Sequence variants possess different affinities to substrates [89] due to surface charge and hydrophobicity dissimilarities. The amino acid sequence of a glycoprotein determines its hydrophobicity, conformation, and charge, amongst other properties [88].

The primary structure (amino acid sequence) of a glycoprotein can determine intermolecular, monomeric association, and aggregation propensity [49]. Even minor sequence differences can cause conformational differences leading to product variants with different biophysical attributes and virus filter fouling propensity. Inadvertent substitution of hydrophilic amino acids with hydrophobic amino acids or vice versa in the polypeptide sequence amplifies sequence variants.

#### 3.4.4. Micro-Heterogeneity-Induced Product Variants

mAbs, antibody fragments, bispecific antibodies, and Fc-fusion proteins are expressed in mammalian cells such as Chinese Hamster Ovary (CHO) cells for pharmaceutically relevant glycosylation profiles [16]. Flynn et al. reported that a typical CHO cell culture batch of mAbs has three major glycan species present, and they are G0F, G1F, and G2F [90]. These three dominant glycan structures are dependent on cell lineage and culture parameters [91]. *E. coli* expresses mostly insoluble, non-glycosylated variants [75]. Hybridomas offer a rapid expression template for initial product manufacture [16,92,93].

During cell culture and harvesting operations, expressed glycoproteins are usually not uniformly glycosylated [38,94,95]. Glycoproteins are expressed with a range of glycosylation profiles depending on cell culture conditions [96,97,98,99,100,101]. Micro-heterogeneity of glycoproteins can occur as a result of differences in glycosylation and other post-translational modifications. Variations in appended glycans introduce charge heterogeneity to the product monomer and determine the glycoprotein’s native fold state, aggregate susceptibility, and stability [102,103,104,105]. These product variants can affect the performance of virus filters.

Glycans are hydrophilic oligosaccharide moieties typically appended to glycoproteins in the cell during glycoprotein synthesis [106]. Glycans assist proper folding of the polypeptide chain before product secretion [74,94,107]. Most therapeutic proteins are glycoproteins. Glycoforms of protein products introduce structural heterogeneity, which affects their affinity to substrates, their stability, and other physicochemical characteristics of these therapeutic proteins [106,108,109]. Even in the same cell culture batch, a range of glycoforms occur [106,110,111]. Glycoforms occur due to skipped glycosylation sites on the glycoprotein or differences in the structure of appended glycans [94].

Glycan type and abundance can alter the product’s biophysical properties. Several studies have looked at the stability of different mAb glycoforms. These results show that aggregation is more prevalent in unglycosylated mAbs since glycans modulate aggregation [106,112]. Furthermore, a study showed that in terms of physical stability between pH 4–6, di-glycosylated IgG1-type mAbs were the most stable, and mono-glycosylated IgG1 was the least stable [113]. Post-translational modification can strongly affect the pI of a glycoprotein [74,106]. Variations in the pIs of product variants can affect hydrophobic and electrostatic interactions.

## 4. Mitigation of Virus Filter Fouling

### 4.1. Prefiltration before Virus Filtration

Even though the support structure of the virus filter can function as an inline prefilter, significant fouling is often observed due to the product- and process-related foulants listed above that could be present in the feed stream. Standard practice involves the inclusion of a virus prefilter to remove these contaminants. Virus prefilters may rely on one or more mechanisms of action for the removal of foulants.

A prefilter, often inline, is added upstream of a virus filter to increase permeate flux and productivity. The improvement in performance depends on the biotherapeutic product properties, prefilter material, and buffer conditions [2]. The mechanisms and conditions for foulant capture are different for different prefilters [2]. Table 4 gives a non-exhaustive list of common prefilters used to capture foulants and mitigate fouling of the virus filter. Size exclusion prefilters such as the 0.1- and 0.22-micron filters remove aggregates larger than the respective size cut-off of the prefilters. Ion exchange prefilters are more effective at low conductivity due to the reduction in electrostatic shielding.

The Sartobind S and Q are commonly used as polishing steps, usually before the virus filter. They are run as a separate unit operation (not inline) and can also provide significant virus clearance by adsorption. Brown et al. indicate that virus filter throughput may be increased by adsorptive ion exchange membrane prefiltration [114]. Consequently, if an ion exchange polishing step is used just before the virus filtration step, it may lead to higher fluxes and productivity during virus filtration. However, if the prefiltration step is not conducted in line, the improvement in performance of the virus filter will depend on the hold time between the two unit operations and the product properties under the buffer conditions of interest.

Wickramasinghe et al. opined that trace amounts of aggregates that have a diameter less than 50 nm play a significant role in virus filtration membrane fouling [2]. These small aggregates with diameters less than 50 nm cannot be removed by 0.1 μm or 0.22 μm size exclusion filters but can block the virus filter pores. Virus filtration membranes typically have a pore size around 20 nm at the separation-active layer. Soluble aggregates (20–50 nm) can be removed using adsorptive prefilters (cation exchange, anion exchange, multimodal) to prevent fouling of virus filters. Adsorptive prefilters have been shown to bind aggregates, thereby reducing subsequent fouling of virus filters. Adsorptive prefilters work well for product oligomers in the 600–1500 kDa range, which cannot be removed by 0.22-µm size exclusion prefilters. Ion exchange prefilters have shown great potential in clearing aggregates for effective downstream processing operations [115].

Endotoxins can be removed using hydrophobic prefilters, which bind the phosphorylated lipid moiety, or using anion exchange prefilters to capture the polysaccharide moiety [69,72,116]. Anion exchange membranes work well for endotoxin removal due to the positively-charged ligands binding with the negatively-charged endotoxin (isoelectric point = 1–4).

Hydrophobic prefilters require a moderate/high salt content (ionic strength) to reduce the product’s solvation layer, enabling exposed hydrophobic patches to adsorb on the hydrophobic prefilter. Hydrophobic interaction prefilters can be effective in removing product variants with different hydrophobicity, as well as some of the more hydrophobic product aggregates.

Ion exchange prefilters are helpful in the downstream removal of HCPs due to the pI difference between mAbs and most HCPs. DNA is strongly negatively charged in aqueous solution and can be effectively removed using anion exchange membranes during polishing operations [20,116].

Multimodal prefilters are useful to filter out foulants that cannot be removed by ion exchange, size exclusion, or hydrophobic interaction-based prefilters alone. These multimodal prefilters include the three prefilters from MilliporeSigma, as shown in Table 4.

### 4.2. Mitigation of Virus Filter Fouling Using Process Parameters

Monoclonal antibody properties are highly dependent on the buffer conditions and excipients that are part of the formulation. Excipients are non-drug substance components of the formulation. During high-throughput screening of mAbs for optimum buffer conditions, a specific buffer type and composition may be found to inhibit aggregation and mitigate fouling of virus filters. Phosphate, acetate, and tris buffers may work for some biomolecules, while viscosity-inhibiting buffers may be preferred for highly concentrated mAbs. Arginine reduces mAb monomeric self-association, non-specific membrane interactions, and mAb aggregation [117]. Excipients such as histidine and arginine help to marginally improve the stability of monomeric species during formulation [35,118]. These excipients can result in cost reduction for virus filter consumables but may require further removal before drug substance delivery to the patient.

## 5. Outlook

Achieving high levels of virus clearance for mammalian cell-derived biotherapeutics will continue to be a challenge. There is a continuing need for better virus filters that maximize productivity, flux, and LRV for batch processes. As biomanufacturing moves towards continuous manufacturing platforms, there will be a need to develop new virus filters. Unlike current virus filters, which are designed to process a product batch in one shift, in continuous biomanufacturing, the virus filter will be run for much longer times and likely at much lower filtrate flux/transmembrane pressure. Further development of virus filters for continuous operations will be needed.

There is a growing demand for virus particles and virus-like particle-based vectors for delivery of gene therapies and vaccines. Virus particle-based delivery systems such as attenuated, recombinant, infectious, and inactivated virus particles, as well as virus-like particles and even subunits of virus particles, are highly effective therapeutics. However, downstream purification of these new therapeutics is challenging. Future virus filter designs will need to be optimized for these emerging therapeutics.

## Figures and Tables

**Figure 1 bioengineering-09-00155-f001:**
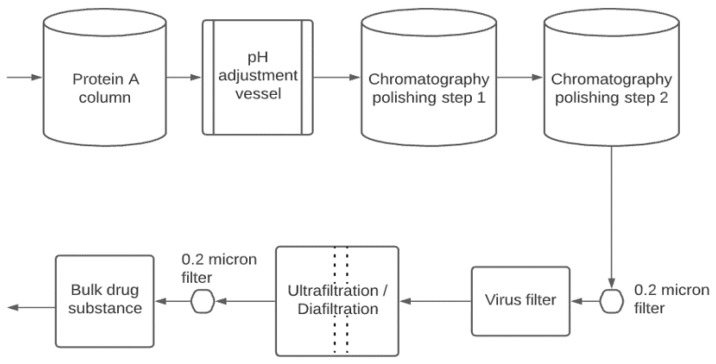
Downstream purification of mammalian cell-derived biotherapeutics.

**Table 1 bioengineering-09-00155-t001:** Examples of approved Chinese Hamster Ovary (CHO) cell-derived biotherapeutics. Non-exhaustive list compiled from publicly available resources (https://www.accessdata.fda.gov/scripts/cder/daf/index.cfm?event=BasicSearch.process), US Food and Drug Administration, (last accessed 10 January 2022), European Medicines Agency [9].

Drug Classification	Examples	First Approval by FDA	Manufacturer
Monoclonal antibodies	Pembrolizumab	2014	Merck
	Nivolumab	2014	Bristol Myers Squibb
	Aducanumab	2021	Biogen
	Avelumab	2017	EMD Serono
	Omalizumab	2003	Genentech
	Adalimumab	2002	Abbvie
	Tezepelumab-ekko	2021	Amgen/AstraZeneca
Fc-fusion proteins	Abatacept	2021	Bristol Myers Squibb
	Aflibercept	2011	Regeneron
	Alefacept	2003	Biogen
	Etanercept	1998	Amgen
	Rilonacept	2008	Regeneron
Cytokines	Darbepoetin alfa	2011	Amgen
	Interferon beta-1a	2003	Biogen
	Epoetin alfa	2011	Amgen
Enzymes	Agalsidase beta	2003	Genzyme
	Human DNase	1993	Genentech
	Laronidase	2003	Biomarin
	Tenecteplase	2000	Genentech
Hormones	Choriogonadotropin alfa	2000	EMD Serono
	Follitropin alfa	2004	EMD Serono
	Osteogenic protein-1	2001	Stryker Biotech
	Thyrotropin alfa	1998	Genzyme

**Table 2 bioengineering-09-00155-t002:** Some common viruses used for validation studies in biomanufacturing [21].

Name of Virus	Diameter (nm)
Animal parvoviruses (non-enveloped DNA viruses, bovine, canine, or porcine)	18–24
Poliovirus (picornavirus, non-enveloped RNA virus)	25–30
Encephalomyocarditis virus (EMC, picornavirus, non-enveloped RNA virus)	25–30
Feline calicivirus (calicivirus, non-enveloped RNA virus)	35–39
Bovine viral diarrhea virus (BVDV, flavivirus, enveloped RNA virus)	40–60
SV40 (simian vacuolating virus 40, polyomavirus, non-enveloped DNA virus)	45–55
Sindbis virus (togavirus, enveloped RNA virus)	60–70
Reovirus (non-enveloped RNA virus)	60–80
Herpes simplex virus (HSV, *Herpesviridae*, enveloped DNA virus)	150
Pseudorabies virus (PRV, *Herpesviridae*, enveloped DNA virus)	120–200

**Table 3 bioengineering-09-00155-t003:** Commercially available virus filters [2,26,27]. Asahi Kasei Bioprocess is a part of the Asahi Kasei Group; MilliporeSigma is a subsidiary of Merck KGaA.

Filter	Manufacturer	Membrane Material	Configuration	Comments
Planova 15 N, 20 N	Asahi Kasei Bioprocess	Regenerated cellulose	Asymmetric single-layer hollow fibers	Parvovirus filter
Planova 35 N	Asahi Kasei Bioprocess	Regenerated cellulose	Asymmetric single-layer hollow fibers	Retrovirus filter
Planova BioEX	Asahi Kasei Bioprocess	Hydrophilized PVDF	Asymmetric single-layer hollow fibers	Parvovirus filter
Viresolve NFR	MilliporeSigma	Polyethersulfone	Asymmetric triple-layer pleated sheets	Retrovirus filter
Viresolve Pro	MilliporeSigma	Polyethersulfone	Asymmetric double-layer flat sheets	Parvovirus filter
Pegasus SV4	Pall Corporation	Hydrophilized PVDF	Symmetric double-layer pleated sheets	Parvovirus filter
Pegasus Prime	Pall Corporation	Polyethersulfone	Pleated sheets	Parvovirus filter
Ultipor VF DV20	Pall Corporation	Hydrophilized PVDF	Symmetric double-layer pleated sheets	Parvovirus filter
Ultipor VF DV50	Pall Corporation	Hydrophilized PVDF	Symmetric double-layer pleated sheets	Retrovirus filter
Virosart HC	Sartorius AG	Polyethersulfone	Asymmetric double-layer pleated sheets	Parvovirus filter
Virosart HF	Sartorius AG	Modified polyethersulfone	Asymmetric single-layer hollow fibers	Parvovirus filter

**Table 4 bioengineering-09-00155-t004:** Commercially available prefilters, modes of action, and manufacturers [27].

Prefilter	Material	Mechanism of Action	Manufacturer
Planova 75 N	Regenerated cellulose	Size exclusion, removal of small aggregates	Asahi Kasei Bioprocess
Bottle top 0.1/0.22 µm	Polyethersulfone	Size exclusion, removal of large aggregates	Multiple
Pegasus Protect	Nylon	Size exclusion, removal of large aggregates	Pall
Sartobind Q	Quaternary ammonium ligands	Anion exchange	Sartorius AG
Sartobind S	Sulfonic acid ligands	Cation exchange	Sartorius AG
Sartobind phenyl	Phenyl ligands	Hydrophobic interaction	Sartorius AG
Viresolve Pro Shield	Surface modified PES	Size exclusion, ion exchange (cation)	MilliporeSigma
Viresolve Pro Shield H	Surface modified PES	Size exclusion, hydrophobic interaction	MilliporeSigma
Viresolve Prefilter	Diatomaceous earth, cellulose fibers, and a cationic imine binder	Cation exchange, size exclusion, hydrophobic interaction, ion exchange	MilliporeSigma

## Data Availability

The data presented in this study are available from the corresponding author upon request.

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
