# Peer review of "Process- and Product-Related Foulants in Virus Filtration"

_bioengineering, 2022, doi:10.3390/bioengineering9040155_

Round 1

Reviewer 1 Report

The work of Isu et al. reviewed concept and process performance in virus filtration. A concise, yet well written review article. Minor comments on the relation between membrane materials and foulants behavior is barely discussed, whether process parameters or membrane materials matters for future virus filtration. Please elaborate.

Reviewer 2 Report

This review covers the virus filtration in the downstream process focusing on the membrane fouling of virus filters by biological foulants derived from mammalian cell-based biotherapeutics production process.  The manuscript is informative for both bioprocess and membrane communities. I recommend publication after minor revision.

1. Explanation or comments on Table 1 and Table 2 in the text are missing.

2. Manufacturer names should be updated. (ex. MilliporeSigma -> Merck Millipore, Sartorius AG or Sartorius)

3. Table 3 should be reorganized by Manufacturer or Material or Target virus.

4. In Table 4, Sarbobind Q and S are membrane chromatography products. Can we use them for prefilter for virus filtration? If so, please add explanation with references.

5. The contents in this review is described at the end of Introduction section. I hope the author can enhance the paragraph and rearrange the sections rather than parallel configuration. I think section 2 and 3 is general information on bioprocess downstream and virus filtration, and section 4 and 5 are about bio-foulants. They could be combined for easy-reading.

6. Section 6 is about the prefiltration for fouling reduction. Can you add a brief review on other anti-fouling techniques (if possible) and actual benefits of them such as flux enhancement, cost reduction, productivity enhancement or LRV improvement?
